# A novel method for cold region streamflow hydrograph separation using GRACE satellite observations

Shusen Wang[1], Junhua Li[1], and Hazen A. J. Russell[2]

[1]Canada Centre for Remote Sensing, Natural Resources Canada, Ottawa, K1A 0E4, Canada
[2]Geological Survey of Canada, Natural Resources Canada, Ottawa, K1A 0E8, Canada

*Correspondence to*: Shusen Wang (Shusen.Wang@Canada.ca)

**Abstract.** Streamflow hydrograph analysis has long been used for separating streamflow into baseflow and surface-runoff components, providing critical information for studies in hydrology, climate and water resources. Issues with established methods include the lack of physics and arbitrary choice of separation parameters, problems in identifying snowmelt runoff, and limitations on watershed size and hydrogeological conditions. In this study, a GRACE-based model was developed to address these weaknesses and improve hydrograph separation. The model is physically based and requires no arbitrary choice of parameters. The new model was compared with six hydrograph separation methods provided with the U.S. Geological Survey Groundwater Toolbox. The results demonstrated improved estimates by the new model particularly in filtering out the bias of snowmelt runoff in baseflow estimate. This new model is specifically suitable for applications over large watersheds which is complementary to the traditional methods that are limited by watershed size. The output from the model also includes estimates for watershed hydraulic conductivity and drainable water storage, which are useful parameters in evaluating aquifer properties, calibrating and validating hydrological and climate models, and assessing regional water resources.

## 1 Introduction

A streamflow hydrograph is the time-series record of streamflow at a gauging site. Streamflow includes baseflow (the longer-term delayed flow from natural water storage such as groundwater discharge from aquifers) and quick flow (or surface runoff, the short-term response to a rainfall event or snow melt). Separating streamflow observed at a gauging site into baseflow and surface runoff helps characterise watershed hydrogeology and understand the water dynamics such as rainfall-runoff relationships and climate change impact on groundwater discharge (van Dijk, 2010; Gao et al., 2015; Rudra et al., 2015; Foks et al., 2019). Information on baseflow and surface runoff is also critical when dealing with a wide range of water-related issues such as flow regulations, water quality, habitat, reservoir design and operation, and hydroelectric power generation (Boulton and Hancock, 2006; Santhi et al., 2008; U.S. Bureau of Reclamation, 2012; Miller et al., 2016; Nuhfer et al., 2017). Streamflow hydrograph analysis has long been used for separating streamflow into baseflow and surface runoff components and can be traced back to Boussinesq (1904) and Maillet (1905). A wide variety of approaches have evolved since then and several reviews have described this development including Hall (1968), Nathan and McMahon (1990), Tallaksen (1995),

Smakhtin (2001), and Rudra et al. (2015). The approaches started with manual separation of the streamflow hydrograph into surface runoff and baseflow. Two commonly used manual methods include base-flow-recession methods, mostly by constructing a master recession curve to represent a watershed's typical recession behaviour (Rorabaugh, 1963; Stewart, 2015), and curve-fitting methods, mostly by identifying specific points in the hydrograph and connecting them via some predefined rule to account for the shape of the curve between these points (Natermann, 1951; Linsley et al., 1982; Chapman, 1999).

Manual approaches are time consuming and inexact and results can be difficult to replicate among investigators. Attempts to automate manual methods with computers allowed fast and convenient baseflow estimation for multiple watersheds with various spatiotemporal scales, and removed some of the subjectivity inherent in the manual approaches (Arnold et al., 1995; Sloto and Crouse, 1996). However, these approaches basically rely on determining the points where baseflow intersects the rising and falling limbs of the surface runoff response, which are essentially arbitrary (Szilagyi and Parlange, 1998). Various

digital filtering techniques with large variations in complexities have also been used for hydrograph separation, but they still suffer from the lack of hydrological basis and the disadvantage of arbitrary choice of separation parameters (Chapman, 1999; Furey and Gupta 2001; Eckhardt 2005; Piggott et al., 2005; Foks et al., 2019; Shao et al., 2020). The results from these approaches often need to be carefully assessed before they are considered to be hydrologically valid. In particular, most of the existing algorithms are developed and tested for rainfall-dominated watersheds, and few studies have examined their suitability

for snowmelt-dominated systems. Applying algorithms and parameters obtained from rainfall-dominated systems to snowmelt-dominated systems could cause large uncertainties (Voutchkova et al., 2019). Indeed, incorrectly identifying snowmelt runoff as groundwater discharge has long been hypothesized but, to the best of our knowledge, no studies have quantified this bias. Another limitation for the existing approaches is that most of them are limited to watersheds size of no more than 1,000-2,000 km$^2$ (Rutledge, 1998). Despite these limitations, traditional hydrograph separation approaches are still widely used because of

the modest data requirements and ease of implementation. Recent improvements in hydrograph separation includes new parameterisation strategies (Pelletier and Andréassian, 2020) and recognition of multiple baseflow components in the streamflow (Curtis et al., 2020; Stoelzle et al., 2020). Nevertheless, since traditional hydrograph separation methods are based on a number of simplifications and assumptions that limit their applicability, previous studies have widely recognised that more effort is required to evaluate the limitations and their effects, and when possible, the methods should be combined with

other methods and data to address these limitations (Hooper and Shoemaker, 1986; Stewart et al., 2007; Rosenberry and LaBaugh, 2008; Miller et al., 2014;).

    The subsurface drainable water storage is a major driver of baseflow for most watersheds with certain hydrogeological settings. Due to the subsurface heterogeneity in soils and aquifers, subsurface water storage over a large spatial domain is difficult to determine using traditional observation methods such as in situ soil moisture sensors and groundwater wells. This poses a

major challenge for studying the water storage-baseflow relationships. The development of the Gravity Recovery and Climate Experiment (GRACE) satellites, which were launched in 2002, has provided opportunity to overcome this challenge. GRACE provides monthly changes in total water storage (TWS) derived from time-variable gravity observations (Tapley et al., 2004). As the first technique for large-scale TWS measurement, GRACE observations have enabled a wide range of novel research

advancing knowledge for water science and water resources. In the area of river flow hydrology, the innovations include applying GRACE data for quantifying watershed-level drainable water storage (DWS) (Wang and Russell, 2016; Tourian et al., 2018; Macedo et al., 2019; Riegger, 2020), estimating snow mass and snowmelt runoff (Wang et al., 2017), characterizing storage-streamflow relationship and climate change impacts (Riegger and Tourian, 2014; Sproles et al., 2015; Wang, 2019), and assessing flood potential (Reager et al., 2014). In particular, Macedo et al. (2019) used empirical approach and GRACE TWS to estimate non-winter season baseflows at 12 gauge locations distributed throughout the Mississippi River basin in US. In contrast, Wang (2019) and Wang et al. (2017) used winter season data to develop GRACE-based baseflow models for cold region watersheds in Canada. Wang (2019) also revealed the dynamic change of watershed hydraulic conductivity with freezing temperature in winter and expanded the foundation for modelling year-round baseflow using GRACE observations. The objective of this paper is to present a novel method for streamflow hydrograph separation using GRACE satellite observations. The method improves hydrograph separation through addressing the aforementioned weaknesses of traditional methods, such as the lack of physics and arbitrary choice of separation parameters, problems in identifying snowmelt runoff, and limitations on watershed size. The model is demonstrated using the streamflow hydrograph measured at the gauge station for the cold region Albany River watershed located in Canada. The results of our approach are compared with those obtained from six widely accepted methods for streamflow hydrograph analysis provided by the U.S. Geological Survey (USGS) Groundwater Toolbox (Barlow et al., 2015). The output from this study also includes watershed hydraulic conductivity and drainable water storage. These parameters are useful in the evaluation of aquifer properties, for input to hydrological and climate models, and for assessment of water resources. The rest of this paper is organized as follows: Section 2 describes the method. Section 3 identifies the study region and datasets. The results are provided in Section 4 and discussed in Section 5, followed by conclusion remarks in Section 6. A brief description of the six USGS hydrograph separation methods and more details on the datasets used in this study are provided in the supplementary materials for this paper.

## 2 Method

Our GRACE-based hydrograph separation method is based on two assumptions. First, the total water storage change of a watershed ($S_{tot}$) is contributed by the changes of (1) surface water ($S_s$) which contributes to surface runoff, (2) subsurface water ($S_g$) which has a delayed discharge and contributes to baseflow and, (3) non-dischargeable water ($S_n$) which makes no contributions either to surface runoff or baseflow. Second, the total streamflow observed at a gauge station ($Q_{obs}$) is composed of surface runoff ($Q_r$) and baseflow ($Q_b$). With the assumptions, we have:

$$S_{tot}(t) = S_s(t) + S_g(t) + S_n(t) \tag{1}$$

$$Q_{obs}(t) = Q_r(t) + Q_b(t) \tag{2}$$

In this study, all the units are in mm (depth of water) for water amount variables ($S_{tot}$, $S_s$, $S_g$, $S_n$), and mm/day for water flow variables ($Q_{obs}$, $Q_r$, $Q_b$), unless specified otherwise. At long-time scales such as monthly ($t$), we ignore the possible desynchrony between storage change and the observed flow which could be due to the water travel time from the site of flow generation to the gauging station. The surface water is treated as being composed of two parts: (1) the part in excess of the surface water retention capacity (i.e., $S_s$), of which the change represents the amount of surface runoff $Q_r$; and (2) the part under the surface water retention capacity, which temporarily stay on the soil surface, infiltrate into the soil at a later date, and finally contributes to baseflow. This part of water is therefore integrated into $S_g$. Water that is non-dischargeable and non-infiltratable such as snow is represented by $S_n$.

The $S_g$ is connected with $Q_b$ by the following baseflow model developed in Wang (2019) for Albany River watershed:

$$Q_b(t) = k(T)\big(S_g(t) - a\big), \tag{3}$$

where
$$k(T) = k_0 \frac{T_c}{T_{acc}(t) + T_c}, \qquad (T_{acc} \leq 0). \tag{3a}$$

The $a$ (mm) in Equation (3) is a parameter representing the threshold value of water storage below which the watershed discharge (or baseflow) would be zero, or above which the water storage is defined as drainable water storage (DWS) which can be regarded as the water in a basin that is connected to streamflow and, with no additional precipitation input, would drain out of the basin as time advances towards infinity (Macedo et al., 2019). This parameter was first introduced to baseflow modelling in Wang et al. (2017). It is necessary to include in constructing the storage-baseflow relationship when using GRACE observations since the GRACE TWS represents the anomaly rather than absolute amount of water storage in a watershed.

The $k$ (day$^{-1}$) is a rate constant, measuring the watershed lump hydraulic conductivity for subsurface water to discharge. The $k$ is commonly regarded as a static parameter being determined by watershed hydrogeological characteristics, such as geomorphology, soil properties, and aquifer settings. For cold region watersheds, Wang (2019) recently found that the $k$ is quite dynamic in the winter season and it can be significantly reduced with freezing conditions. In this study, the $k$ is estimated using Equation (3a) as proposed in Wang (2019), where $T_{acc}$ (°C day) is the accumulated daily air temperature ($T$) from the start of winter and it is reset to 0 when the winter season is over. The starting and ending dates for a winter season are estimated using several criteria based on temperature and they are detailed in Wang (2019). The $T_{acc}$ represents the accumulated coldness at a specific time in winter and it is used as a proxy for freezing conditions. It has the advantage of being simple and easy to obtain. The $k_0$ in Equation (3a) is the base value of $k$, or the $k$ when soil frost is not present. The $T_c$ (°C day) is a parameter and when $T_{acc}$ reaches the value of $T_c$, the $k$ will be reduced by half. The relationship is supported by results from process-based land surface model simulations for soil ice content variations with accumulated temperature in winter (see Fig. S1 in Wang, 2019).

The baseflow model contains three unknown parameters: $k_0$, $T_c$, and $a$. For cold region watersheds in winter with frozen soil and snow-covered ground surface, the observed streamflow is solely contributed by baseflow so that $Q_b=Q_{obs}$. With this advantage, the winter season data were used for the calibration of the baseflow model and the solution of these parameters. A triple-nested numerical iteration scheme was developed to find the optimum values of the $k_0$, $T_c$, and $a$ for achieving the maximum Nash-Sutcliffe efficiency (NSE) between observed and modelled baseflows. The initial values for the $k_0$, $T_c$, and $a$ can be estimated from either available information or expert knowledge. In this study, the values obtained in Wang (2019) were used.

With the assumptions of Equations (1) and (2), and the baseflow model of Equation (3), the baseflow contribution to the total streamflow can be obtained analytically:

$$Q_b(t) = \frac{k_0 T_c [S_{tot}(t) - \Sigma Q_{obs}(t) - S_n(t) - a]}{T_{acc}(t) + T_c - k_0 T_c}. \tag{4}$$

Surface runoff is then calculated as the difference between total streamflow and the estimated baseflow. The estimation for baseflow and surface runoff is made for their monthly values, constrained mainly by the monthly temporal resolution of GRACE data. The $\Sigma Q_{obs}(t)$ in Equation (4) thus represents the sum of observed daily streamflow in a month.

The six methods for streamflow hydrograph separation provided with the USGS Groundwater Toolbox (Barlow et al., 2015) include PART, HYSEP (Fixed Interval, Sliding Interval, and Local Minimum), and BFI (Standard and Modified). A brief description for each of the methods is provided in the supplementary material. More details can be found in Barlow et al. (2015) and Sloto and Crouse (1996). Results from these six USGS methods and our approach were compared, with a focus on seasonal variations and long-term averages in the estimated baseflow and baseflow index (BFI).

## 3 Study Region and Datasets

The streamflow hydrograph measured at the mouth of the Albany River was used for the model demonstration. The Albany River is located in the Far North of Ontario, Canada, predominantly between 49 and 52 degrees north (Fig. 1). It has a length of 982 kilometres and a drainage area of 137,230 km$^2$. The river flows northeast from Lake St. Joseph at an elevation of 371 metres into James Bay. The headwater of the watershed is situated in the Canadian Shield physiographic region which is characterized by a thin soil layer over Precambrian bedrock and moderate topographic relief. The middle and lower portions of the watershed are within the Hudson Bay Lowlands (HBL) physiographic region which is characterized by Paleozoic bedrock overlain by glacial sediment and poorly drained organic deposits with low topographic relief. Within the study area, the Canadian Shield landscape is dominated by Boreal forest which transitions into Barren Boreal and Taiga vegetation zones within the HBL. The Albany River watershed is highly vulnerable to flooding in spring due to snowmelt and sensitive to climate change (McLaughlin and Webster, 2014).

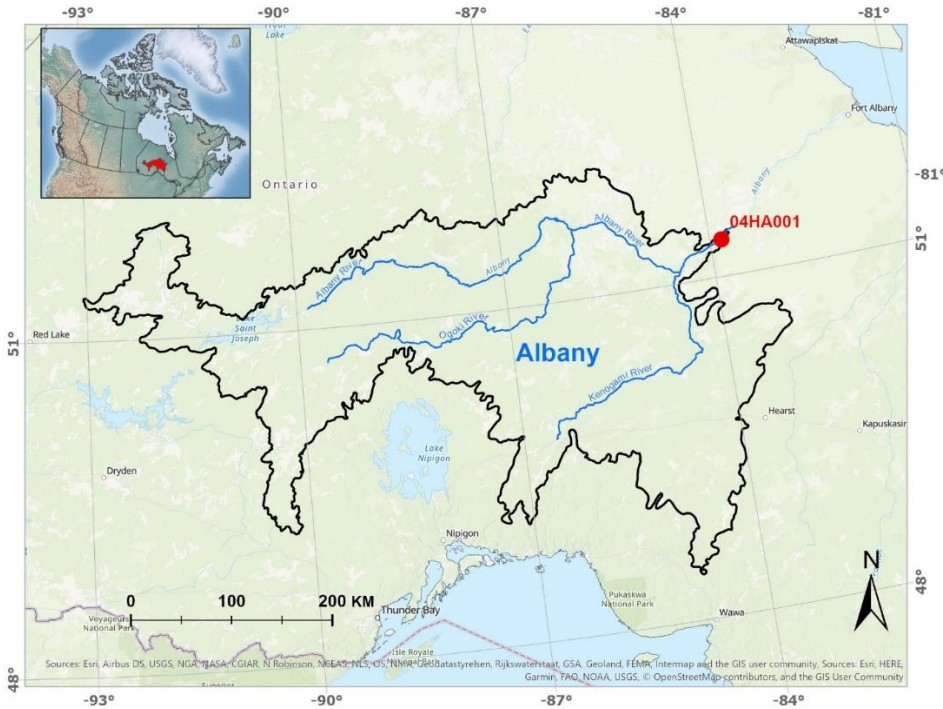

Figure 1. Location and geometry of the Albany River watershed and the gauge station. Study area in red on inset of Canada
160                 (© OpenStreetMap contributors 2020. Distributed under a Creative Commons BY-SA License).

Four main datasets were used in this study, which include air temperature ($T$) from the Global Land Data Assimilation System (GLDAS) V2 meteorological forcing, snow water equivalent ($S_n$) from the land surface model EALCO (Ecological Assimilation of Land and Climate Observations, Natural Resources Canada) V.4.2, river flow measurement at gauging station
04HA001 ($Q_{obs}$), and total water storage change ($S_{tot}$) from GRACE Release-06 V03 spherical harmonic (SH) solutions. Details of the datasets and their quality evaluations are given in the supplementary material. The study time period covers 15 years of 2002–2016. The watershed hydroclimatic conditions characterised for this period from these datasets are summarised below. The watershed has a cold, humid climate. During the study period the watershed had a mean annual temperature of 1.0°C. Daily temperature dropped below 0°C on average in late October to early November, and rose above 0°C in middle April of
the next year (Fig. 2). Both of the transitional periods have large inter-annual variations of more than a month. The lowest air temperature in the study period was colder than -25°C in late January. The extremely low temperature and low solar radiation in the winter season resulted in deep-frozen condition for the watershed in winter, which minimized the possible contribution of surface runoff to streamflow. Also, the long winter season commonly exceeded five months of each year and provided a relatively large amount of data for baseflow model calibration.


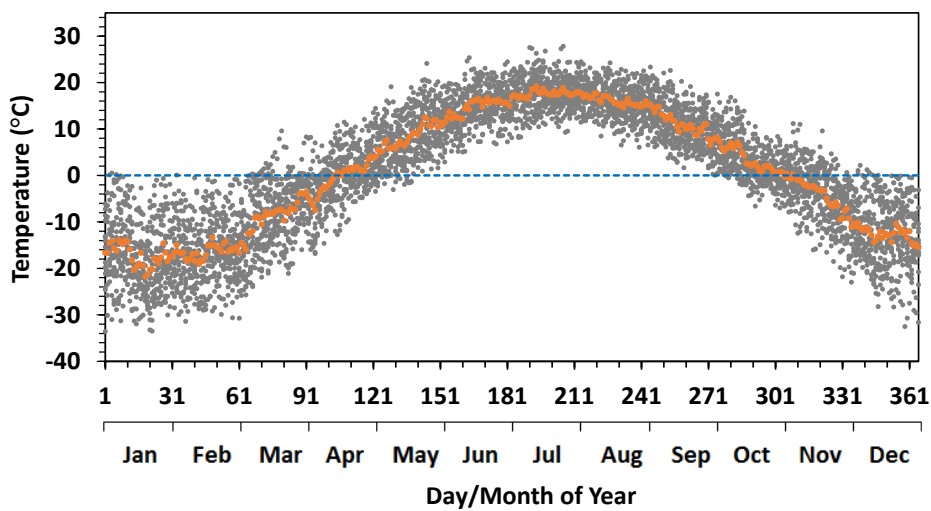

Figure 2. Daily air temperature for the Albany River watershed over the study period (2002-2016). Orange dots represent the 15-year mean values.

The watershed has a large water budget surplus for aquifer recharge and to sustain year-round river flow. Annual precipitation for the study period averaged 784 mm (Fig. 3), more than twice the annual evapotranspiration of 340 mm (Wang et al., 2013). Precipitation in summer (rain) accounted for 71% of the total annual precipitation. The rain intensity was mostly under 20 mm/day, which is fairly low, and consequently large rain induced streamflow peaks are uncommon. The rest of the precipitation occurred in winter as snow, with an intensity rarely over 10 mm/day. Snow mostly accumulated from freeze-up

till the melt season in the spring (Fig. 4).

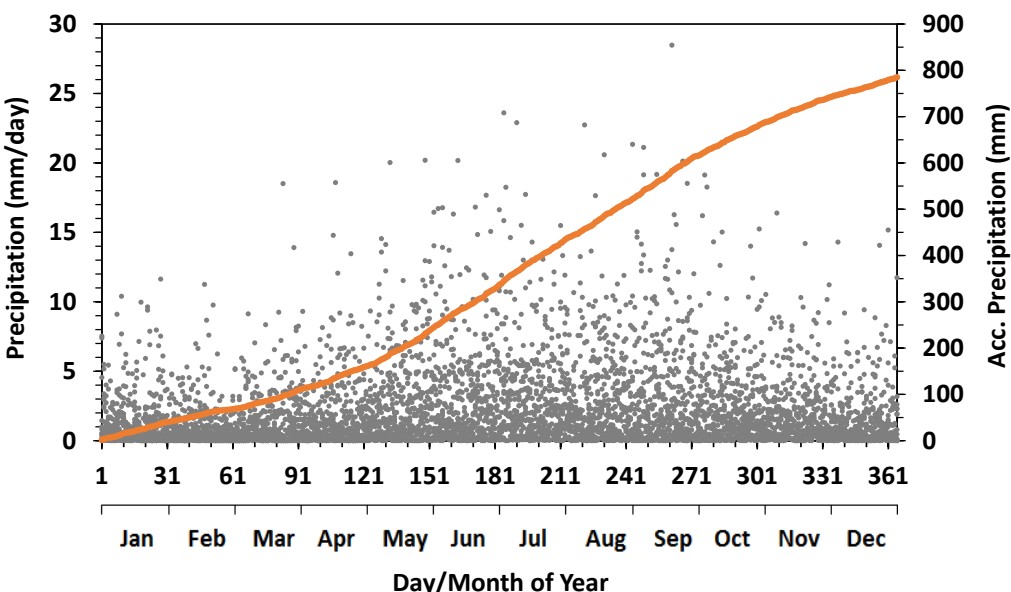

Figure 3. Daily precipitation for the Albany River watershed over the study period (2002-2016). Orange line represents the accumulated precipitation in a year using the 15-year mean daily values.


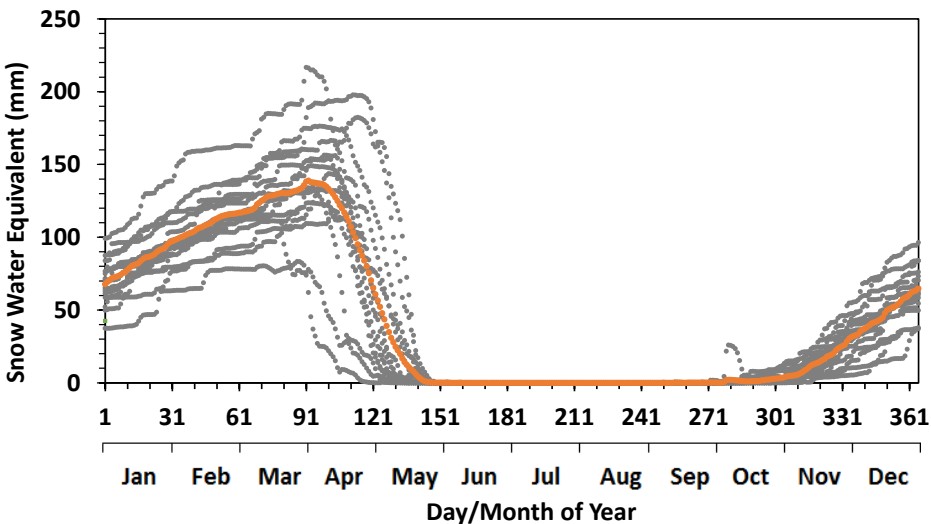

Figure 4. Daily snow water equivalent for the Albany River watershed simulated by the EALCO model over the study period (2002-2016). Orange dots represent the 15-year mean values.


The Albany River had an annual mean flow, during the study period, of 1420 m³/s (1.04 mm day⁻¹), with an annual total of 44.8 km³ (380 mm). The peak flow for each year occurs mostly in May due to snowmelt (Fig. 5). The largest peak flow occurred in 2006 and was 8000 m³/s (5.86 mm day⁻¹). Summer flows have minor-peaks from rain events and occurred sporadically throughout the season. They were generally smaller than the spring snowmelt peak. The 15-year (2002-2016)

average flows showed a pattern of sharp decrease from May to August, and then an increase in early autumn till freeze-up. River flow in the winter season decreased smoothly with time and it presented a typical baseflow recession process, confirming our assumption of absent surface runoff due to the frozen conditions. The inter-annual differences in the baseflow values were large in early winter, with a range from over 4600 m³/s (3.37 mm day⁻¹) to just 350 m³/s (0.26 mm day⁻¹) depending on the pre-winter water conditions of the watershed. After discharging over an entire winter season, and before the start of the snowmelt

season, the river had flow values within a small range of 100-200 m³/s (0.07- 0.15 mm day⁻¹). The mean river flow for the winter season varied from 200 m³/s to 700 m³/s (0.15-0.51 mm day⁻¹), with an overall average of 420 m³/s (0.30 mm day⁻¹).

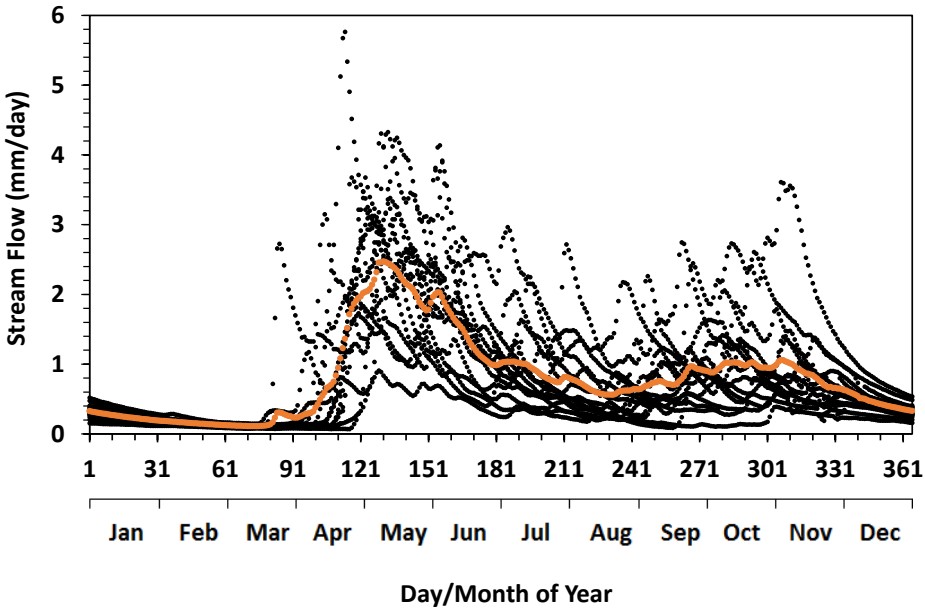

Figure 5. Daily streamflow measured at the gauge station 04HA001 (see Fig. 1) of the Albany River over the study period
(2002-2016). Orange dots represent the 15-year mean values.

The watershed had a maximum variation for total water storage of 200 mm during the 15-year study period (Fig. 6). The lowest TWS values appeared in September and the highest values occurred in April before snowmelt began. Obviously, the increase in TWS in the fall-winter season is mainly due to the snow accumulation and low water loss from snow sublimation, and the

decrease in TWS in the spring-summer season is mainly due to the large amount of discharge of snowmelt water and high evapotranspiration which reaches about 80 mm for land surface evapotranspiration and 130 mm for water surface evaporation

in July (Wang et al., 2014a). The inter-annual variations in TWS for a specific month was large, with a range of 100 mm on average. Note that the baseline (reference) value of the TWS, which is the average over the period of January 2004 to December 2009 in the original datasets, is adjusted to the minimum value found over the study period in this study.


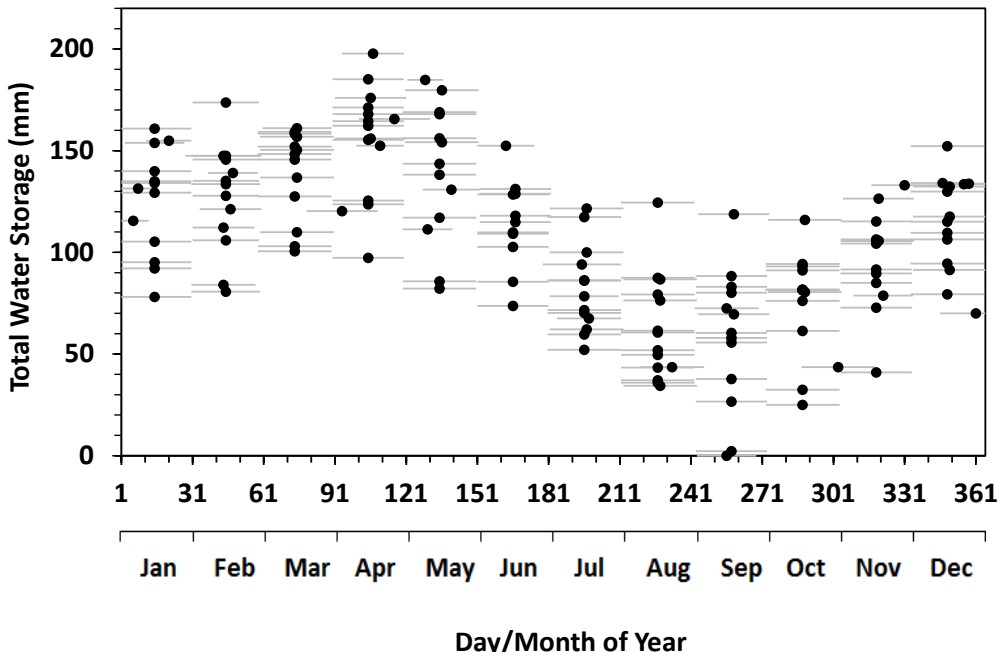

Figure 6. Monthly total water storage (TWS) for the Albany River watershed over the study period (2002-2016, relative to the minimum value for the period). The grey lines represent the time period of GRACE observations used for deriving the TWS.


## 4 Results

The baseflow model calibration results and performance evaluation (Table 1) show that the modelled and observed monthly baseflow values in the winter seasons of 2002-2016 achieved a Pearson correlation coefficient ($R$) of 0.91 ($p<0.001$). The Nash–Sutcliffe efficiency coefficient (NSE) of the model reached 0.823. The model suggested that the watershed had a $k_0$ for subsurface water discharge of $7.45\times10^{-3}$ day$^{-1}$. The $k$ was reduced by half when the accumulated freezing temperature reached -595 °C·day ($T_c$) in winter. When the watershed reached its lowest water storage during the 15 years, which was observed in September 2006, the watershed still had 45.7 mm (5.4 km$^3$) of water available for discharge ($a$) (Fig. 7). On average, the model suggested that the DWS of the watershed was 152.4 mm (18 km$^3$) during the 15-year study period. The seasonal variation of DWS, from its lowest monthly mean value of 103.6 mm (12.2 km$^3$) in September to its highest value of 196.3 mm (23.2 km$^3$)


in April, exceeded over 60% of its average value. The interannual variation of DWS was also large. The largest monthly interannual variation appeared in September, with a value of 118.7 mm (14 km³), and the lowest interannual variations appeared in March, with a value of 60.5 mm (7.1 km³). The overall annual variation range was 87.0 mm (10.3 km³).

Table 1. Baseflow model calibration and test results.

| Model | Parameter | Description | Value |
|---|---|---|---|
| $Q(t) =$ $k_0 \dfrac{T_c}{T_{acc}(t) + T_c}(S(t) - a)$ | $k_0$ (×10⁻³ day⁻¹) | Baseflow rate constant | 7.45 |
| | $T_c$ (°C·day) | Parameter for impact of freezing temperature in winter on $k_0$ | -595 |
| | $a$ (mm) | Drainable water storage (DWS) threshold value, relative to the minimum $S_{tot}$ observed during the study period | -45.7 |
| Model Performance | $MAE$ (mm day⁻¹) | Mean absolute error | 0.05 |
| | $R$ | Pearson correlation coefficient | 0.91 |
| | $NSE$ | Nash–Sutcliffe model efficiency | 0.823 |
| | $p$ | Significance level | <0.001 |


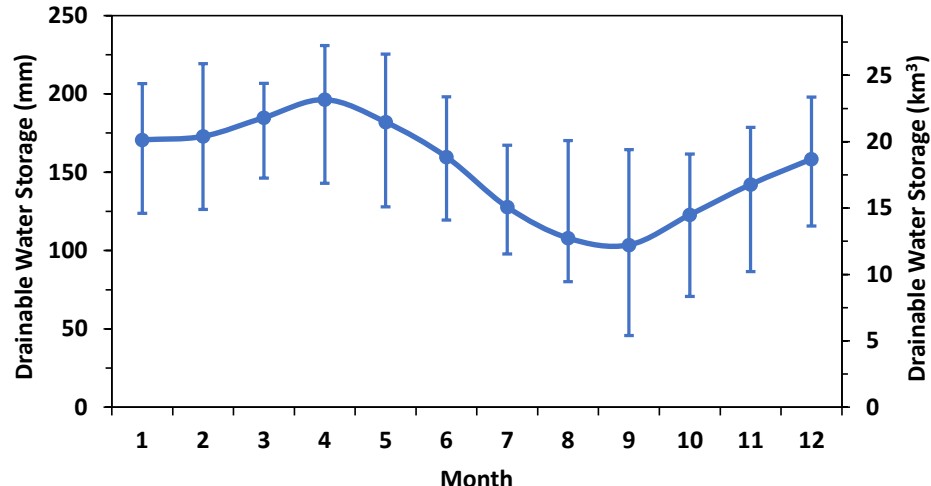

Figure 7. Drainable water storage (DWS) estimated for the Albany River watershed. The blue line represents the mean values, and the vertical bars represent the range of variations for the 15-year of 2002-2016.


The baseflow hydrograph estimated by our model is shown in Fig. 8(a), compared with the corresponding results obtained from PART (Fig 8b), HYSEP (Fixed Interval, Sliding Interval, and Local Minimum) (Fig 8c), and BFI (Standard and Modified) (Fig 8d). The monthly means over the 15-year study period are compared in Fig. 9, with the corresponding Base Flow Index (BFI) values (or the ratio of baseflow to total streamflow) shown in Fig. 10. Overall, the baseflow and BFI estimated from these seven methods showed general agreement. Their similarities and differences can be generalised by the following three time periods.

(1)   Winter Season. We used fairly strict criteria (Section 2) to select the winter months to ensure that the watershed was in frozen conditions and no rain and other events contributed to river flow by surface runoff. The PART, BFI-Std and BFI-Mod models basically showed BFI of 1.0 in mid-winter. With the assumption of baseflow as the only contribution to streamflow in the winter months, our model has a BFI value of 1.0 in December, January and February. This is consistent with the observation of the streamflow data in Fig. 5 which exhibited typical baseflow recession process (Fig. 5) during these periods. The temperature during these three winter months in each year of our study period all had values significantly below 0°C. The six USGS methods also showed similar results. In particular, PART and the two BFI methods (Standard and Modified) estimated baseflow basically being equal to the total streamflow in January and February, but they estimated small but noticeable contributions of surface runoff in December (4% for PART, 9% for BFI-Standard and Modified). The three HYSEP methods showed relatively large difference to our model. They all estimated surface runoff contributions in the streamflow in each of the three winter months. The results from the HYSEP Fixed and Slide methods were virtually the same, which showed BFI values increasing from 0.88 in December to 0.92 in January and 0.93 in February. In contrast, the HYSEP Minimum method showed different results from the other two HYSEP methods in December (BFI=0.77) and February (BFI=0.97). Close examinations of the data revealed no hydrological processes (e.g., rain or snowmelt from above 0 temperature) that could lead to surface runoff generation during these time periods, and the results from the three HYSEP methods are likely due to the defects with the algorithms of these methods and the observation noise. Despite the differences in BFI values among some of the methods as discussed above, they had little impact on the annual total baseflow estimates since the overall winter streamflow was very low (Fig. 5).

(2)   Snowmelt Season. In the spring snowmelt season, total streamflow increased sharply to its annual peak in May. The baseflow estimated from our model also reached its annual peak at this time, representing 54% of the total streamflow (Fig. 10). In the rising limb of the streamflow peak, our model obtained its lowest BFI value in a year, which was 0.44 for April. This is consistent with the fact that in the early snowmelt season, the soil was in a frozen condition which prevented water from infiltrating the surface, resulting in snowmelt water mostly contributing to the streamflow as surface runoff. At the same time, aquifer discharge (baseflow) remained the lowest in the year. In the falling limb of the spring streamflow peak, our model estimated a much smaller decreasing rate in baseflow than the total streamflow (Fig. 9). The modelled BFI value for June was 0.69, a significant increase from its values in April (0.44) and May (0.54). The six USGS methods also showed similar patterns in the baseflow and BFI variations, but in some cases with substantial differences. Specifically, all six methods also obtained their highest baseflow values during the peak flow time in May (Fig. 9), their lowest BFI values in

the rising limb in April (Fig. 10), and continuous increase in BFI values through the snowmelt season (Fig. 10). The large

decrease in BFI from February to April (Fig. 10) reflects the snowmelt runoff contribution to the streamflow. Overall, our

model showed the lowest baseflow and BFI values among the seven methods during the snowmelt season. It is worth noting

that the baseflow/BFI values for April from our model was close to those from four of the USGS methods, namely HYSEP

Minimum, the two BFI methods and PART. However, this better agreement in model results may not suggest more robust

estimate in baseflow, which will be discussed later. Among the six USGS methods, the most striking difference was the high

baseflow/BFI values by the HYSEP Fixed and Slide methods throughout the snowmelt season. For example, the BFI

estimated by these two methods was about 0.64 for the month of April, while the estimates from the other five methods

including our model were around 0.42. The BFI-Modified method estimated a relatively high baseflow/BFI value for the

peak flow month of May, and it is worth mention that this is the only month of the year that the BFI-Modified method

showed significantly different result from the BFI-Standard method. Another noticeable outlier is from PART in the falling

limp (June), which showed a baseflow/BFI value similar to that in the peak flow time of May.

(3)  Summer Season. The results estimated by the seven methods were fairly close in summer season (Fig. 9). The

general pattern can be characterized by the low baseflow in August–September when the watershed had the lowest water

storage, and a second peak around October before the winter season starts. In terms of BFI, all the methods estimated

relatively high BFI values in August, with the highest value estimated by our model (Fig. 10). During the second peak in

October, our model showed a large decrease of BFI (0.76) from the summer low-flow in August (BFI=0.96), suggesting the

increased contribution of surface runoff from rain to the streamflow. In contrast, the decrease of BFI during this transitional

period was not obvious for five other models except HYSEP Minimum.

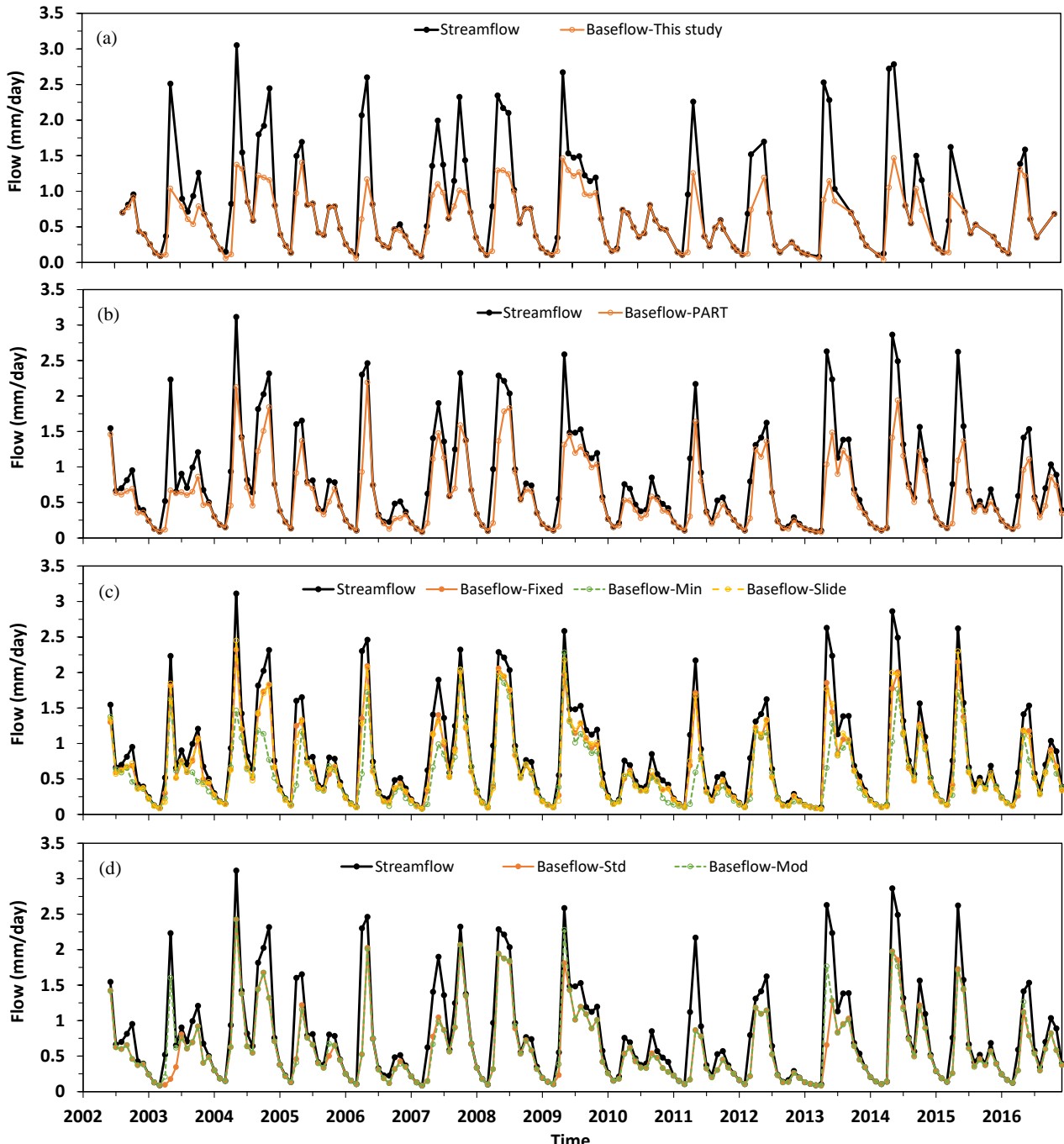


Figure 8. Hydrograph separation of the Albany River for 2002-2016. Data shown are monthly values for the observed total streamflow (black line/dot in each panel) and the baseflow estimated by the methods of (a) this study, (b) PART, (c) HYSEP, including Fixed Interval, Sliding Interval, and Local Minimum, and (d) BFI, including Standard and Modified. Note that the

months in (a) were based on GRACE monthly data temporal coverage (see Fig. 6) which varied over the years and were slightly different from the calendar months used for the other models shown in (b), (c) and (d).

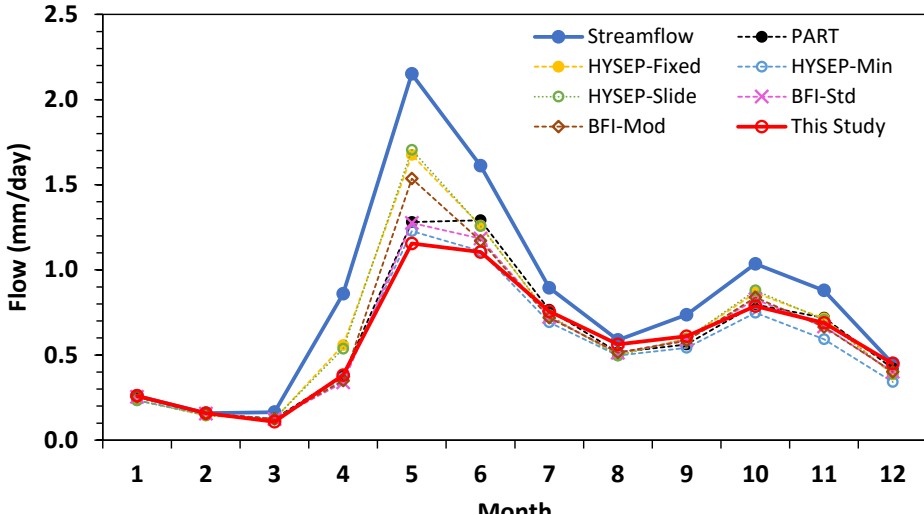

Figure 9. Observed total streamflow and estimated baseflow by different methods for the Albany River. Data shown are monthly means in the 15 years of 2002-2016.


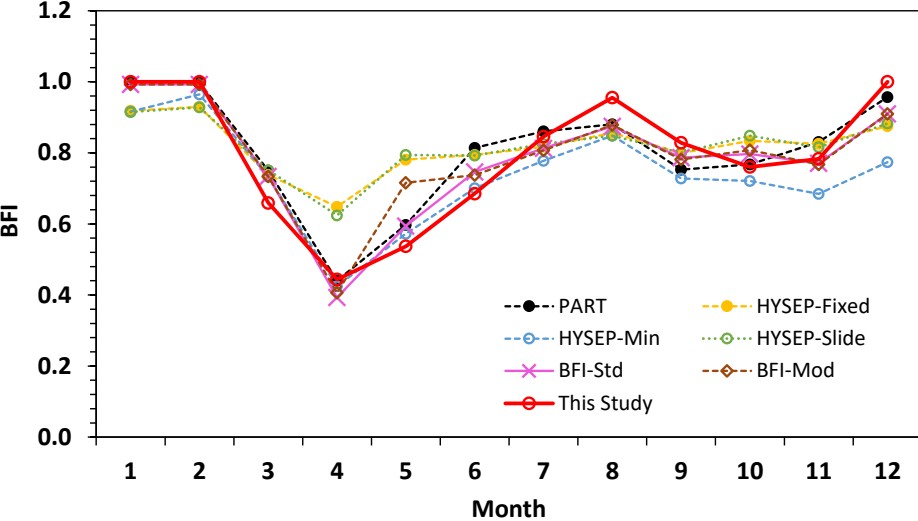

Figure 10. Baseflow index (BFI) estimated by different methods for the Albany River. Data shown are calculated using the 15-year (2002-2016) means of baseflow and streamflow.

Overall, our model showed relatively larger seasonal variations in BFI compared with the six USGS methods (Fig. 10). The contribution of baseflow to the total streamflow estimated by our model was 71.7% for the 15 years. In comparison, the corresponding estimates by the six USGS methods ranged from 67.6% (HYSEP-Minimum) to 79.7% (HYSEP-Slide) (Table 2). The PART method showed the best agreement with our model, with a correlation coefficient of R=0.92 for the monthly baseflow estimates (Fig. 11). The difference between the results from HYSEP-Fixed and HYSEP-Slide was very small, but

they differed significantly with the HYSEP-Minimum method. The HYSEP-Minimum method estimated lower baseflow than the other two HYSEP methods systematically year-round, and the difference was especially significant in the peak flow season and early winter. The BFI-Standard and Modified methods obtained very similar results, except in three months during the 15 years when the Modified methods estimated much higher baseflow than the Standard method, and all three exceptions appeared in May, the month with peak streamflow.


Table 2. Average baseflow and baseflow index (BFI) estimated for 2002-2016.

| Method | This study | PART | HYSEP | | | BFI | |
|---|---|---|---|---|---|---|---|
| | | | Fixed | Minimum | Slide | Standard | Modified |
| Baseflow (mm/day) | 0.585 | 0.605 | 0.649 | 0.552 | 0.650 | 0.587 | 0.609 |
| BFI | 0.717 | 0.742 | 0.795 | 0.676 | 0.797 | 0.719 | 0.746 |

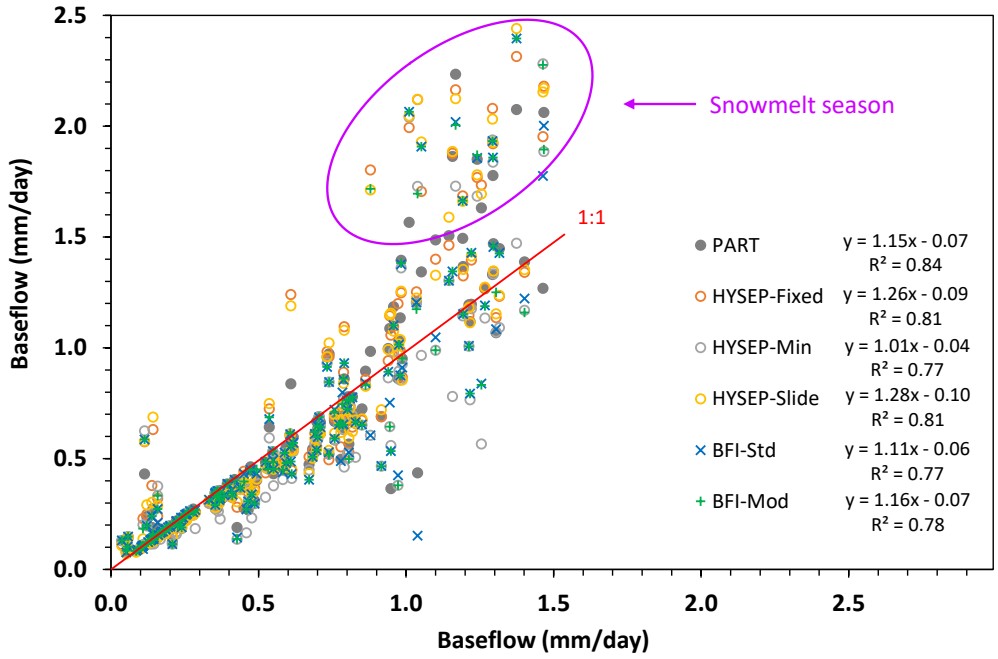

Figure 11. Comparisons of baseflow estimated from this study (X-axis) with those from PART, HYSEP (Fixed Interval, Sliding Interval, and Local Minimum), and BFI (Standard and Modified) (Y-axis).

## 5 Discussion

Snowmelt is a known hydrologic process that could cause problems with the traditional hydrograph separation methods.

Snowmelt in high latitudes can be a slow process which lasts for weeks depending on the rising temperature trend of the spring season. The hydrograph fluctuations caused by snowmelt runoff have very different characteristics from those caused by rain events. The algorithms for the traditional hydrograph separation methods may incorrectly identify streamflow increase contributed by snowmelt runoff as groundwater discharge (Barlow et al., 2015). Indeed, we found that the baseflow values from the six USGS methods during the snowmelt season were all higher than that from our model, suggesting overestimation

of baseflow due to inclusion of snowmelt runoff. This difference is particularly large with the HYSEP Fixed and Slide methods (Fig. 9). Our model demonstrates improvement to this known bias in these traditional models, and provides the most conservative estimate of baseflow. Our data and results did not suggest systematic bias coming from the calibration process. First, the range of drainable water storage used for model calibration reasonably covered the variation range during the summer season. Second, our model results reversed the relationship of baseflow when compared with other models in summer. In fact,

our model obtained higher baseflow in the mid-summer months of July–September (Figure 9) than most of the other six models. Annually, our model showed baseflow close to BFI-Standard, higher than HYSEP-Minimum, and lower than the other 4 models (Table 2). Third, our results are in good agreement with, and supported by, the results obtained in other studies, including Rudra et al. (2015) which analysed BFI for 115 Ontario watersheds, and Wang and Russell (2016) and Wang et al. (2017) which explicitly estimated the snowmelt runoff.

It is worth noting that among the six USGS methods, the HYSEP Minimum method provided the lowest estimate of baseflow almost year-round in all seasons. Similar results were also report in Curtis et al. (2020) where the six USGS methods were applied to 312 watersheds in the USA for baseflow separation. They found that the HYSEP Minimum method provided the lowest estimate of baseflow, and thereafter suggested HYSEP Minimum as the most robust method for removing snowmelt runoff in baseflow estimates. From our analyses, we found that the conservative estimate of baseflow during the snowmelt

season by the HYSEP Minimum method is likely due to the systematic underestimation of baseflow in its hydrograph separation algorithm.

In the early snowmelt season of April before the streamflow peak, our model estimated similar baseflow values to four of the six USGS methods, in contrast to its consistent lower values for March, May, and June than all of the USGS methods. Where there is better agreement in model results, it does not necessarily mean there is a more robust estimate. On the contrary, it is

reasonable to believe that this could be due to the overestimation of baseflow by our model for April. One assumption of our model is that all the drainable liquid water storage under the surface water holding capacity belongs to the subsurface water

storage and contributes to watershed discharge or river baseflow. In reality, in the early snowmelt season, the soil is still frozen, which will prevent the snowmelt water from infiltrating the surface and contributing to baseflow. This often results in surface water puddles in spring as commonly observed in cold regions. Our above model assumption could be challenged under this condition and it will lead to the overestimation of baseflow.

Large size and low relief of a watershed are also the factors of caution when using the traditional hydrograph separation methods. Rutledge (1998) recommended a maximum drainage area of about 1,300 km$^2$ for the application of these methods. The low relief of a watershed may also affect the time period of surface runoff that can be determined by Equation (A1), and this effect could be exacerbated by the large drainage area of a watershed. Halford and Mayer (2000) and Halford (2008) also questioned the use of Equation A1 to determine the duration of surface runoff for large watersheds. In contrast, the algorithm of our model is not limited by Equation A1, rather it is limited by the GRACE footprint which is over 10$^5$ km$^2$. In this study, we used the Albany River watershed for demonstration which has a total drainage area of over 137×10$^3$ km$^2$, or two orders of magnitude larger than that suggested by traditional methods. The method could be applied to small watersheds when high-resolution information is available for water storage.

The Albany watershed in this study was among the watersheds with the smallest water imbalance (non-closure) in Canada (Wang et al., 2014a; Wang et al., 2014b). This suggests high data quality over this region, including for GRACE TWS, EALCO snow water equivalent and river flow observations, and close connections between surface water and groundwater systems for the Albany watershed. The uncertainties with snow (Sn), surface water (Ss), and subsurface water (Sg) cannot be directly evaluated because no corresponding observations are available. However, they can be evaluated indirectly. We calculated the GRACE TWS measurement error, leakage error, and combined total error following Wahr et al. (2006) using the land surface model CLM4, and they were 13.2 mm, 15.8 mm, and 20.6 mm, respectively, for the watershed. The impact on the TWS error estimate due to the uncertainty in the CLM4 model was evaluated by comparing with the error estimate using a different land surface model of NOAH (Wang et al., 2014a). The magnitudes of errors from the two studies were found to be similar. Since our model is calibrated using observed baseflow measurement, systematic errors or biases in TWS would be reflected in the model calibration process and compensated in the parameter values, so their impact on the hydrograph separation results would be minimal. Random errors in TWS could directly affect the baseflow estimation. However, in the cold season since the TWS change is mainly due to snow variations which does not contribute to river flow, the uncertainties in TWS also have minor impact on baseflow estimate. In the non-frozen season, an error of 20 mm in TWS would result in an estimate error of 0.15 mm/day in baseflow, which is small compared with the flow magnitude in summer of the watershed. In fact, as suggested by the water budget closure study and error analyses (Wang et al., 2014a; Wang et al., 2015), the random error is much smaller than the measurement error of 20mm. The small water imbalance found for the Albany watershed also suggests that the impact of possible inter-watershed groundwater flows on the modelling was small. However, it is worth noting that for watersheds with significant inter-watershed groundwater flows (e.g., Bouaziz et al., 2018; Hulsman et al., 2021), neglecting this process may lead to misrepresentation of the natural hydrological system by the model which could cause system biases.

The uncertainties with snow can be further evaluated by comparing this study with Wang (2019), where the winter time drainable water was estimated by the difference of the TWS at the winter starting time and the accumulated streamflow (from winter start to a specific month). The advantage of Wang (2019) approach is that it doesn't need snow data in the modelling, while the disadvantage is that it increased the impact of the uncertainties of winter streamflow measurement in the modelling. Also, any errors with the TWS at the winter start will propagate into the modelling for all the months in that season. The

modelling results show that this study, which used snow data, slightly improved that of Wang (2019) (NSE=0.823, r=0.91 vs. NSE=0.809; r=0.903), suggesting the impact of uncertainties from EALCO snow data is less than that from the uncertainties in streamflow measurements on the baseflow modelling.

The model was calibrated using all the available winter data during our study period. Even so, due to the limited GRACE observations and the strict selection criteria for winter months to ensure that the watershed was in frozen conditions and no

rain or other events could occur to cause surface runoff, the sample number for model calibration was relatively small (~3 months/year on average). However, the calibration doesn't assume that the baseflow mechanism is stable over the years. In fact, possible variations in baseflow mechanism with watershed conditions (such as yearly weather changes) were expected, and the impact, which causes deviations between modelled vs. observed flows, was reflected in the calibration results. Moreover, the model treated the baseflow coefficient dynamically in winter using the accumulated freezing temperature

function to account for the seasonal (e.g., winter vs. summer) and yearly changes in watershed conditions. This represents a major innovation for this method, and the improvement and advantage over traditional methods of treating baseflow parameters as constants have been discussed in detail in Wang (2019). In summer, our model is basically a modified linear reservoir model and the baseflow coefficient is treated as a constant. It is interesting to note that our model estimated little surface runoff, or high baseflow contribution in streamflow, for the years with dry summer (e.g., 2005,2006,2010,2011) (Fig. 8). This is not due

to high baseflow coefficients in these summers. Rather, this is the result that our model estimated the watershed water storage was mainly attributed to subsurface water in these time periods.

This study provides a technical framework for hydrograph separation using GRACE observations. It is demonstrated using streamflow measurement for a cold region watershed. Cold region watersheds in winter have frozen soil and snow-covered ground surface, so the observed streamflow is solely contributed by baseflow. This provides an advantage for the calibration

of the baseflow model. Moreover, by using only-winter data the model calibration reduces the impact of a number of hydrological processes on the solution of model parameters, such as evapotranspiration, soil surface infiltration and groundwater recharge. For applications of our model over other climate regions, the model parameters could be estimated by different approaches, such as using dry season data over arid regions when rainfall-induced surface runoff is absent in streamflow measurements.

It is worth noting that the Albany groundwater discharge is mainly contributed by surficial aquifers and the baseflow-storage relationship is relatively stable (Wang 2019). The hydrogeological settings of the watershed are relatively simple  and the baseflow mechanism can be very different for watersheds that have steep slopes, complicated aquifer systems, permafrost layers, or large variations in water conditions, which often involve dynamic channel networks (e.g., time-varying geometry of

saturated channeled sites) and result in complicated storage-baseflow relationships (Biswal and Kumar, 2014; Biswal &

Marani, 2010; Streletskiy et al., 2015; Tallaksen, 1995). Indeed, the study of Wang (2019) tested different baseflow models and found that the Albany watershed follows a simple linear baseflow-storage relationship quite well when soil frost is absent (i.e., summer season), and the freezing temperature function (Eq. 3) allows the model to account for the dynamic changes in watershed winter condition. For watersheds with complicated hydrogeological settings, our approach may need to involve more comprehensive baseflow models or TWS-dependent dynamic parameters to address the complicated aquifer systems and

flow mechanisms.

## 6 Conclusions

A GRACE-based hydrograph separation model is developed in this study to address the weaknesses of traditional methods in baseflow estimate, such as the lack of physics and arbitrary choice of separation parameters, problems in identifying snowmelt runoff, and limitations on watershed size and other conditions. The model first constructs a baseflow model using winter data,

and then uses streamflow observations to solve the baseflow in all seasons. It is physically-based and does not require arbitrary choice of parameters. The hydrograph separation results from our model can generally be characterized by the absence of surface runoff in winter when the watershed was covered by snow and in frozen conditions, high volume but low percentage contribution from baseflow to the streamflow in the snowmelt season, and low volume but high percentage contribution from baseflow to the streamflow in the summer dry season. Comparisons of the model with the six hydrograph separation methods

provided with the U.S. Geological Survey Groundwater Toolbox show that our model effectively filtered out the snowmelt runoff from baseflow estimate. The algorithm of the model is not limited by watershed size and it is specifically suitable for applications over large watersheds using GRACE observations, which is complementary to the traditional methods that are mostly limited to smaller watersheds. The model output also includes estimates for watershed lump hydraulic conductivity and the drainable water storage, which are useful parameters in evaluating aquifer properties, calibrating and validating

hydrological and climate models, and assessing regional water resources.

*Data availability.* Data used in this study are openly available and can be downloaded from the following links: air temperature: http://mirador.gsfc.nasa.gov/, snow water equivalent: ftp://ftp.ccrs.nrcan.gc.ca/ad/EMS/EALCO/, river flow: https://wateroffice.ec.gc.ca/, total water storage change https://podaac.jpl.nasa.gov/GRACE.


*Author contributions.* S.W. conceived and implemented the research. S.W. wrote the initial version of the paper. J.L. contributed Fig. 1. All authors contributed to discussion and improving the paper. H.A.J.R. contributed project managerial support.

*Acknowledgements*. This study was supported by the Cumulative Effects Project and Climate Change Geoscience Program of the Natural Resources Canada.

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
