# Peer review of "A novel method for cold region streamflow hydrograph separation using GRACE satellite observations"

_Hydrology and Earth System Sciences, 2020_

## Referee Comment (RC1) · Anonymous Referee #1 · 18 Dec 2020

This paper proposes a new method to separate the baseflow from the total runoff using GRACE observations. An analytical relation was derived for the baseflow with three unknown parameters. This relation was derived with the assumption that the surface runoff is zero in the winter such that the total discharge is equal to the baseflow for these months. This new method was tested in a snow-dominated region and compared to several existing methods focusing on the baseflow hydrograph and baseflow index for the winter, snowmelt and summer seasons.

This paper is well written and structured. Also, it is an interesting topic which could contribute to improved baseflow separation in large river basins. I have a couple of

major and minor comments that would improve the paper.

Generic comments:

1. When deriving the analytical solution (Eq.4), it seems the following assumption was made: $S_s = 0$ with $S_s$ the surface water storage. However, in the winter when the surface runoff is equal to zero, the surface storage is not necessarily zero. Please clarify this as this could influence the validity of Eq.4. Also, for months with very low TWS values (e.g. $S_{tot} \approx$ -100 mm in September according to Figure 6), the baseflow $Q_b$ seems to be negative according to Eq.4 which does not make sense and is not visible in Figure 8. So what did you do for these months when $S_{tot} < a$?

2. The resulting baseflow according to the proposed method depends on the parameter values mentioned in Table 1. However, it remains unclear how these parameters were estimated. In line 126, it is mentioned the model was calibrated with an iteration scheme, but additional details are missing. For example, which scheme was applied, how many different parameter combinations were tested and what parameter ranges were used? It would be interesting and valuable to look into different parameter combinations with similar performances and how this could influence the results. How much do you think your results would change if you would use a different model performance metric or calibration scheme?

3. According to Figure 8, the baseflow is relatively high compared to the total discharge. Do you have any observations that could verify this? Is it possible all methods significantly overestimate the baseflow especially during months when the fast runoff is not zero? Would you get similar results with the manual approach?

4. Please include in the Discussion a section on GRACE uncertainties and how this could affect your results. Based on Figure 1, there are several open water bodies in and near the basin which could affect the GRACE observations. Also, please discuss whether this methodology is applicable in other regions that are not snow-dominated. For example, areas with zero rainfall during dry seasons in arid regions? Are there
specific criteria (climate, minimum/maximum basin size etc.) for which this methodology is applicable? Do you have any recommendations to further improve this approach and maybe expand it to other regions with different climatic conditions?

Specific comments:

1. Line 11: Please specify the region for which your approach is valid.

2. Line 11: Please specify what you mean with "the model [. . .] does not require a priori parameterisation" as there are three unknown parameters which were calibrated (line 121).

3. Line 21: What about groundwater flow from deep aquifers that contribute to the baseflow?

4. Line 22: What about rapid subsurface flow that contribute to fast runoff but where the water is located below the surface?

5. Line 23-26: Please explain more detailed how baseflow separation helps with respect to hydrogeology characterisation, rainfall-runoff relationships, flow regulations, water quality etc.

6. Line 20 – 26: Please add references to support this section.

7. Line 29: Are there no recent reviews on this topic?

8. Line 30: Please explain the manual approach more detailed. Based on which criteria are baseflow and fast runoff typically distinguished?

9. Line 42: What causes large uncertainties in snow-dominated regions?

10. Line 45: Why is there a limit on the watershed size?

11. Line 72: Please be more specific: which weaknesses do you intend to address with your new method?

12. Line 93: Do you mean dS/dt = Q on monthly time scale? Are you referring to the

total storage (Stot) and total discharge (Qobs)?

13. Line 96: Do you mean dSs/dt = Qr?

14. Line 97: Are you assuming all water stored on the surface infiltrates into the soil and hence contributes to the baseflow? Is this always the case?

15. Line 111: Are you sure k is the hydraulic conductivity and not the reservoir coefficient which indicates the mean residence time in the basin and depends on the hydraulic conductivity as well as other properties?

16. Line 126: Based on Table 1, you also used the Pearson correlation coefficient, significance level and mean absolute error to evaluate your results. Please mention that here.

17. Line 135: Please explain how you compared the different methods. For example, you used the baseflow index (BFI) and specifically looked at three seasons (winter, snowmelt and summer season).

18. Section 3: Please clarify clearly which months belong to which season (summer, winter, autumn, snowmelt etc.). Based on which criteria did you identify the winter period (fixed months, based on the temperature or something else)?

19. Line 174: How much is the annual evaporation?

20. Line 193 "less than half of the spring snowmelt peak": It doesn't look like less than half in Figure 5.

21. Line 207: maximum variation of what exactly?

22. Line 211: How high does the evaporation get in this region where the average temperature is 1°C? In other words, how significant is the contribution of the evaporation?

23. Line 226: How did you estimate the drainable water storage? Please explain this in Section 2.

24. Line 258: Please be more specific: how and based on which criteria did you examine the data?

25. Line 261 "little impact": Please show some numbers to support this statement.

26. Line 310 – 315: Are you referring to Figure 11 here? If yes, your observations are very difficult to see in that figure.

27. Data availability: There is no data available for the snow water equivalent with the given link and the link for the discharge data is not valid anymore.

28. Supplements: Please specify the version of the temperature and snow data used in this study.

Technical comments:

1. Line 13 and 14: "estimates" instead of "estimate"

2. Figures 2, 4 and 5: Please add the vertical blue lines shown in Figure 3 to allow easier comparisons.

3. Figure 5: Please show the start of the winter in this figure.

4. Line 225: Please refer to Figure 7 here.

5. Figure 7: Please add the variable name in the label of y-axis.

6. Figures 8-10: Please mark the following seasons in this figure: winter season, snowmelt season and summer season. You compare the different techniques for these seasons, hence marking them would allow for a better graphical comparison for specific seasons. Also, please use the same colours in Figure 8 as in Figures 9 and 10.

7. Line 265: Please refer to Figure 10 here.

8. Line 269: Please illustrate this in a figure.

9. Line 272: Please refer to Figure 9 here.

10. Figure 11: Is your model on the x- or y-axis? Please use the same colours as in Figure 10. It would also be helpful if you would add a 1:1 line.

11. Lines 327, 328, 332: Please remove "may"

12. Line 338: Please replace "recommend" in this sentence.

13. Line 352: Please remove "accurately"

---

## Referee Comment (RC2) · Anonymous Referee #2 · 8 Feb 2021

General comments: The research on baseflow separation has a long history and there have been a variety of methods as summarized by Pelletier and Andreassian (2020). This study proposed a new method for snow-dominated regions that bases on watershed-scale water balance approach. Apart from streamflow observations, it relies on the water storage change observations from GRACE, air temperature and the watershed snow water equivalent observation from assimilated dataset. Through comparing with the other six hydrograph separation methods, the proposed method provides the most conservative estimate of baseflow in light of consideration of snowmelt runoff. However, the method is built on some strong assumptions, which must be consolidated by further evidence. Overall, the manuscript is well written. But the following issues

should be addressed properly before the paper can be considered for publication in the HESS. Thus, the paper needs a major revision.

Major comments: 1. As one strong assumption of the physically-based method, the watershed water balance closure can be captured by the suggested observations. First of all, the surface water budget could be closed with detailed observations, but the groundwater water system may not be closed in the watershed. Secondly, the authors did not provide the uncertainties of all components, particularly the non-dischargeable water change (Sn). Thus, more discussion is needed for this assumption. 2. Another implicit assumption is the invariant model parameters (k0, Tc, and a). If my understanding is correct, the authors only derived the three values from the model calibration shown in Table 1. I am not sure if they use the 15-year data for the calibration. If so, this means the baseflow mechanism is stable over the years. In fact, the baseflow mechanism could be influenced by yearly weather changes. If not, please provide the variation of the yearly calibrated parameters. Furthermore, in light of the contrast hydrological mechanisms between the winter season and the summer season, the assumption of the invariant model parameters within a hydrological year might be also problematic. More evidence is needed for this assumption. 3. The authors declare the proposed method yields the most conservative estimates of baseflow in comparison with other methods. However, since the model parameters are derived from the winter season, systematic bias might come from here when using them for the snowmelt season and the summer season. 4. This novel method was only applied in one watershed, so the applicability for other large size watershed was not examined. And the conclusion that filtering out snowmelt runoff bias in baseflow estimates was not strictly tested and quite occasional, it could be the possibility that this model underestimated baseflow. One possible reason is that using the single winter data to calibrate parameters, in fact, the runoff components in winter were quite different from those in spring/summer.

Minor comments: 1. Line 54-55: Is baseflow primarily driven by the subsurface drainable water storage in all situation watersheds? Please be more specific. 2. Line 86 and

Line 97: Is surface water which is under the surface water holding capacity contribute to baseflow, in some cases it could turn into non-dischargeable water. 3. Line 126: The nested numerical iteration scheme was not clarified. 4. Line 130: The dimensions were inconsistent in Eq. 4, where Stot [mm] vs Qobs [mm/day]. Please clarify this. 5. Line 211: high evapotranspiration, please quantifying it. 6. Line 246: Why did you artificially set and thought that BFI is 1.0 in the winter season, while all other models estimated that surface flow also existed in winter. Please find these paper: Streletskiy, D.A., Tananaev, N.I., Opel, T., Shiklomanov, N.I., Nyland, K.E., Streletskaya, I.D., Tokarev, I., Shiklomanov, A.I., 2015. Permafrost hydrology in changing climatic conditions: Seasonal variability of stable isotope composition in rivers in discontinuous permafrost. Environ. Res. Lett. 10, 95003. https://doi.org/10.1088/1748-9326/10/9/095003. 7. Please discuss more quantitively the advantages and disadvantages of your model compared with others. 8. Line 330-335: Lack of logic. Not convinced. 9. Line 351: any progress in recent hydrograph separation applied in large size watersheds? 10. Line 355-356: No support information.

---

## Author Comment (AC1) · 25 Feb 2021

We greatly appreciate the comments which we believe will help improve our manuscript. Our responses to the comments are listed below.

Generic comments: 1. Eq.4 does NOT assume Ss=0. It treats Ss as being composed of two parts: the part above the surface water retention capacity, which contributes to surface runoff; and (2) the part below the surface water retention capacity, which is integrated with subsurface water through soil surface infiltration and contributes to baseflow. So, when the surface runoff is equal to zero, the surface storage is not necessarily zero (but is below the surface water retention capacity, e.g., for a lake

or pond without overflow), regardless in which season of the year. The text for this description is revised to make it clearer (the 2nd paragraph of Section 2). The baseline of the TWS data, which was based on the 2004-2009 average in the original data, was readjusted to the minimum value that occurred over the study period. This information was originally given in the Supporting Information (with more details in Wang, 2019), and it is added to the main text in this revision (including re-plot of Fig 6).

2. The manuscript is revised by adding details for the parameter estimation. The numerical scheme for this study, coded in FORTRAN, will be made available to the public through the Canada Centre for Remote Sensing Open Data Portal. The program includes a triple-nested numerical iteration for the three parameters of k0, Tc, and a. We used an iteration of 50 for each parameter with its given range of values. So, the total parameter combinations for a numerical experiment is 50^3. This takes about 10 CPU-minutes for a regular desktop computer. The initial values for the parameters can be estimated from either available information or expert knowledge. In this study, the values obtained in Wang (2019) were referenced. Prior numerical experiments with relatively large ranges and coarse resolutions for the parameter values could be helpful to narrow down the final ranges. The solution with maximum Nash-Sutcliffe Efficiency corresponds to a unique combination of k0, Tc, and a, and the solutions with a range of top Nash-Sutcliffe Efficiency values include a large number of combinations of k0, Tc, and a. This can be seen from the Figure RC1-1 attached. Numerically, it reflects the interactions of parameters in the baseflow simulations (e.g, low a compensates low k0). Physically, it suggests the level of accuracy for parameters with the prescribed accuracy of Nash-Sutcliffe Efficiency. As such, the large number of parameter combinations is an over exaggeration, as many of them are under the prescribed accuracy level.

3. Our method has relatively low annual BFI value compared with five of the six USGS methods which have known problems of overestimating baseflow by including snowmelt runoff (Table 2). Our model may overestimate baseflow in early snowmelt season (April) due to the impact of frozen soil which is discussed in the 3rd paragraph

of Section 5. The high BFI in summer could be attributed to the large size, flat to-pography, and well drained soil (e.g., peatland) of the Albany watershed. Rudra et al. (2015) analysed BFI for 115 Ontario watersheds (note that some of them are very small in size). Our results showed similar magnitudes to Rudra's study for the comparable watersheds. In general, the available data and studies don't suggest baseflow over-estimation by our model (except, possibly in April, as discussed). This will be further verified when more information becomes available.

4. Section 5 is revised to address the comments, including discussion on basin size (paragraph 4), impact of GRACE TWS uncertainties (paragraph 5), applications over different climate regions and further research recommendations (paragraph 6), as copied below. "The GRACE TWS measurement error, leakage error, and combined total error were calculated following Wahr et al. (2006), and they were 13.2 mm, 15.8 m, and 20.6 mm, respectively, for the watershed. The impact of the TWS errors and other uncertainties on our results is generally small. Since our model is calibrated us-ing observed baseflow measurement, systematic errors or biases in TWS would be reflected in the model calibration process and compensated in the parameter values, so their impact on the hydrograph separation results would be minimal. Random errors in TWS could directly affect the baseflow estimation. However, in cold season since the TWS change is mainly due to snow variations which doesn't contribute to river flow, the uncertainties in TWS also have minor impact on baseflow estimate. In non-frozen season, an error of 20 mm in TWS would result in an estimate error of 0.15 mm/day in baseflow, which is generally small compared with the overall flow magnitude for the watershed. This study provides a technical framework for hydrograph separation using GRACE observations. It is demonstrated using streamflow measurement for a cold region watershed. Cold region watersheds in winter are with frozen soil and snow-covered ground surface, so the observed streamflow is solely contributed by baseflow. This provides an advantage for the calibration of the baseflow model. Moreover, by using only-winter data the model calibration reduces the impact of a number of hydro-logical processes on the solution of model parameters, such as evapotranspiration, soil

surface infiltration and groundwater recharge. For applications of our model over other climate regions, the model parameters could be estimated by different approaches, such as using dry season data over arid regions when rainfall-induced surface runoff is absent in streamflow measurements. It is worth noting that the Albany groundwater discharge is mainly contributed by surficial aquifers and the baseflow-storage relationship is relatively simple. For watersheds with complicated hydrogeological settings, the approach may need to involve more comprehensive baseflow models or TWS-dependent dynamic parameters to address the complicated aquifer systems."

Specific comments: 1. The approach is illustrated using a cold region watershed, but it is not limited to cold regions. This is discussed in Paragraph 6 in Section 5. 2. Revised. 3. Revised. More discussion is given in Paragraph 6 in Section 5. 4. Baseflow vs. surface runoff, particularly for large watersheds, is somewhat conceptual and not strictly differentiable. There are studies towards the recognition of multiple baseflow components in the streamflow. We included discussions and cited relevant studies in the paper. 5, 6 and 7. Detailed examples for the baseflow/surface runoff applications in various areas and more recent review works are provided by citing available studies. 8. Revised by providing more detailed information for the manual approach and by adding references. 9. One process is that, as discussed in the following text, incorrectly identifying snowmelt runoff as groundwater discharge with the traditional methods. Addressing this problem is one of the major research objectives of this study. 10. It is due to the assumptions for these methods, for example, the estimation of duration of surface runoff (Eq. A1 in Supporting material file). 11. Revised by adding specifics, which actually are discussed in detail in the 2nd paragraph of the Introduction. 12 and 13. Revised to make the statement clear. For each month, the change of surface water above retention capacity is surface runoff, and the change of Sg contributes to baseflow. Sn has no contribution to either. 14. No. Please see clarification above. 15. A review of literatures apparently show that this parameter is called in many different ways. We revised it to rate constant to avoid any misinterpretations. Its reverse represents residence or turnover time, both are controlled by aquifer properties. 16. The

Nash-Sutcliffe modelling Efficiency is used as the criteria in model calibration. Other statistical parameters are used for model evaluation. 17. Revised. 18. As mentioned in our paper, the interannual variations of temperature is huge. The determination of, for example, winter or snowmelt season, is by temperature-based multi-criteria for each individual year and they varied by calendar months from year to year. Manuscript was revised by adding the information. 19. Specific number (340mm) is added. 20. Revised. From long-term statistical mean view point (Fig. 9), this is the case. 21. Revised. TWS varied from the lowest (0, the reference value) to the maximum of about 200mm. 22. Reference added. As shown in Fig 12 and 13 in Wang et al. (2013), the annual ET over the region is mainly contributed by the three months of Jun-Aug. 23. It is described in the paragraph below Eq. (3a) in Section 2. 24. Revised. 25. Fig.5 was cited. As the orange line shows, the winter season Q contribution to the annual Q is very small. The specific magnitudes were discussed in the paragraph above the figure. 26. Reference to Fig 10 is added. The discussion here is a synthesis of Fig 10, Table 2 and Fig 11. 27. Updated. 28. Revised.

Technical comments: All comments are addressed. Note that the blue lines in Fig. 3 were removed. The transition dates varied from year to year. The blue lines may cause confusion. Given the same reason, adding average transition dates to other figures may cause confusion. We used the same color legend in Fig 9 and 10. Fig. 8 has more colour choice as the models are separated in 4 panels, and we used the same color order among the 4 panels. We used lighter color in Fig. 11 than Fig 10 as there is a lot more overlaps in Fig. 11 than Fig 10.

Reference: Rudra, R., Ahmed, I., Khan, A. A., Singh, K. G., Goel, P. K., Khayer, M., and Dickinson, T.: Use of Baseflow Indices to Delineate Baseflow Dominated and Rapid Response Flow Dominated Watersheds. Canadian Biosystems Engineering 57, 1–11. doi.org/10.7451/CBE.2015.57.1.1, 2015.

524, 2020.

[Figure]

[Figure]

Figure RC1-1: Solutions for $k_0$, $T_c$, and $a$ using NSE as the criteria. Ranges of parameter values: $0.006 < k_0 < 0.009$, $430 < T_c < 770$, $36 < a < 55$.

**Fig. 1.**

---

## Author Comment (AC2) · 25 Feb 2021

We greatly appreciate the comments which we believe will help improve our manuscript. Our responses to the comments are listed below.

Major comments: 1. The water balance closures for Canada's watersheds were studied at both short time (monthly) (Wang et al., 2014a) and long-term (30 years) (Wang, et al., 2014b) scales. The Albany watershed in this study was among the best ones with the smallest water imbalance (non-closure) in Canada. In particular with the precipitation product from Canada's meteorological station network, the water balance for the Albany watershed achieved the least water imbalance (close to 0) among all the watersheds (Fig. 11 in Wang et al., 2014a). This suggests the high data qualities over this region (including snow, or the Sn which is discussed in more detail below) and the close connections of the surface water and groundwater systems for the Albany watershed. Regardless, this study doesn't directly involve water balance assumption. It is based on the fact that the GRACE measured TWS is the sum of surface water+subsurface water+snow. The watershed doesn't have glaciers and other water components to significantly contribute to the GRACE TWS.

The uncertainties with snow (Sn), surface water (Ss), and subsurface water (Sg) were mainly from the EALCO model and the GRACE TWS errors. These uncertainties cannot be directly evaluated because no corresponding observations are available. However, they can be evaluated indirectly. We calculated the GRACE TWS measurement error, leakage error, and combined total error following Wahr et al. (2006) using the land surface model CLM4, and they were 13.2 mm, 15.8 m, and 20.6 mm, respectively, for the watershed. The impact on the TWS error estimate due to the uncertainty in the CLM4 model was evaluated by comparing with the error estimate using a different land surface model of NOAH (Wang et al., 2014a). The magnitudes of errors from the two studies were found to be similar. Since our model is calibrated using observed baseflow measurement, systematic errors or biases in TWS would be reflected in the model calibration process and compensated in the parameter values, so their impact on the hydrograph separation results would be minimal. Random errors in TWS could directly affect the baseflow estimation. However, in cold season since the TWS change is mainly due to snow variations which doesn't contribute to river flow, the uncertainties in TWS also have minor impact on baseflow estimate. In non-frozen season, an error of 20 mm in TWS would result in an estimate error of 0.15 mm/day in baseflow, which is substantially small compared with the flow magnitude in summer of the watershed. In fact, as suggested by the water budget closure study (Wang et al., 2014a), the random error is much smaller than the measurement error of 20mm.

The uncertainties with snow can be further evaluated by comparing this study with

Wang (2019). In Wang (2019), the winter time drainable water was estimated by the difference of the TWS at the winter starting time and the accumulated streamflow (from winter start to a specific month). The advantage of Wang (2019) approach is that it doesn't need snow data in the modelling, while the disadvantage is that it brings the uncertainties of winter streamflow measurement into the modelling. Also, any errors with the TWS at the winter start will propagate into the modelling for all the months in that season. The modelling results show that this study (NSE=0.823, r=0.91) slightly improved that of Wang (2019) (NSE=0.809; r=0.903), suggesting the impact of uncertainties from EALCO snow data is less than that from the uncertainties in streamflow measurements on the baseflow modelling.

2. The model was calibrated using all the available winter data during our study period. Even so, due to the limited GRACE observations and the strict selection criteria for winter months to ensure that the watershed was in frozen conditions and no rain or other events could occur to cause surface runoff, the sample number for model calibration was relatively small (∼3 months/year on average). However, the calibration doesn't assume that the baseflow mechanism is stable over the years. In fact, possible variations in baseflow mechanism with watershed conditions (such as yearly weather changes as commented) were expected, and the impact, which causes deviations between modelled vs. observed flows, was reflected in the calibration results. Moreover, the model treated baseflow coefficient dynamically using the accumulated freezing temperature function to account for the seasonal (e.g., winter vs. summer) and yearly changes in watershed conditions. This represents a major innovation for this method, and the improvement and advantage over traditional methods of treating baseflow parameters as constants have been discussed in detail in Wang (2019). Specifically for the Albany watershed, it had a large water budget surplus for soil/aquifer recharge and the yearly water condition changes were small (Wang et al., 2014a). Moreover, it is located in a flat region and the baseflow is mainly contributed by surficial aquifers including wetlands. The hydrogeological settings of the watershed are relatively simple and the baseflow mechanism is largely different with the watersheds that have deep slopes and complicated aquifer systems with large variations in water conditions, which often involve dynamic channel networks (e.g., time‐varying geometry of saturated channeled sites) and result in nonlinear storage‐baseflow relationships. Indeed, the study of Wang (2019) found that the Albany watershed follows a simple linear baseflow-storage relationship quite well when soil frost is absent (i.e., summer season), and the freezing temperature function (Eq. 3) can effectively make the model account for the watershed condition dynamic changes in winter. The Fig. RC2-1 attached shows the yearly calibrated baseflow parameter k from our model vs. that inversely calculated from observations. The monthly and interannual variations of k were mainly contributed by the variations in watershed frozen conditions (Wang 2019). After the temperature function modification, the k has a correlation coefficient of r=0.86, demonstrating the impact of interannual variations from other flow mechanisms is small.

3. Our model only yielded more conservative estimates of baseflow during the snowmelt season. The bias of overestimating baseflow in snowmelt season by the other six models due to the inclusion of snowmelt runoff is a well-known issue. Our results demonstrated improvement to this known bias in these traditional models. Further, our data and results didn't suggest systematic bias coming from the calibration process. First, as shown in the Fig. RC2-2 attached, the range of drainable water storage used for model calibration was in 20-150 mm. It covered the variation range during the summer season fairly well. Second, our model results didn't show systematic underestimates of baseflow in summer when compared with other models (Figure 9). In fact, our model obtained higher baseflow in the mid-summer months of July-September (Figure 9) than most of the other six models. Annually, our model showed baseflow close to BFI-Standard, higher than HYSEP-Minimum, and lower than the other 4 models (Table 2). Third, our results are in good agreement with, and supported by, the results obtained in other studies, including Rudra et al. (2015) which analysed BFI for 115 Ontario watersheds, and Wang and Russell (2016) and Wang et al. (2017) which explicitly estimated the snowmelt runoff. The comment of Reviewer#1 also doesn't suggest this systematic bias.

4. We greatly appreciate the comment. Unlike many other hydrograph studies which have large freedom in selecting study watersheds and taking the advantage of long-term (say decades) gauge measurements, the method in this study relies on GRACE data. Due to the coarse resolution (>100,000 km²) and data availability of GRACE observations, the selection of study regions and time periods are largely constrained. In addition, our method is very new. To our knowledge, this is the first time someone has proposed a physically-based hydrograph separation model using GRACE observations. So it is important to have a robust testbed with high quality data. For example, if there are influences of glaciers or hydropower dams in flow measurements, they will affect the hypothesis tests. Large watersheds in western Canada mostly originated from the Rocky Mountains involving permanent snow/glacier influence (Wang et al., 2015), and watersheds in east Canada are mostly under the footprint of GRACE. This is the major reason for using the Albany Watershed in the Hudson Bay region for this study. We investigated other watersheds in central-east Canada. The large watersheds without much disturbances are mainly in the Hudson Bay region. Specifically, there are a total of nine large watersheds as shown in the Fig. RC2-3 attached Table RC2-1 below. Unfortunately, none of them except Albany has complete gauge measurement during the GRACE period of 2002-2016. Also, their sizes are all small and can hardly meet the requirement of GRACE resolution. So, including them would bring critical issues on (1) short data records for model fitting and (2) large uncertainties in GRACE data for under-footprint watersheds. As such, we prefer to focus this paper on presenting the innovative approach, and leave its applications (or possibly further improvement) to the worldwide community for other regions. See Reply #2 and #3 for the rest of this comment.

Minor comments: 1. The sentence is revised to "The subsurface drainable water storage is one of the major drivers of baseflow for most watersheds with certain hydro-geological settings.". 2. Yes, we agree that in some cases the surface water which is under the surface water holding capacity could turn into non-dischargeable water. Since GRACE measures the changes of water, water that stays in the system and is

not contributing to runoff or baseflow is not accounted for in the model. The statement will be revised to be more specific. 3. More details in the numerical solutions will be added. The programming code will be made available to the public through the Canada Centre for Remote Sensing Open Data Portal. 4. Thank for this note. Q should be the monthly sum of daily Q, with unit of mm. 5. The July monthly land surface evapotranspiration is about 80 mm, and water surface evaporation is about 130 mm. The annual total evapotranspiration is about 350 mm, and total water surface evaporation is about 560 mm. Detailed information can be found in Wang et al. (2014a). 6. We used fairly strict criteria (Section 2) to select the winter months to ensure that the watershed was in frozen conditions and no rain and other events were present to cause surface runoff. The PART, BFI-Std and BFI-Mod models also showed BFI of 1.0 in mid-winter. Note that the water infiltrated into soil surface and later outflowed into rivers (regardless of pathways, e.g., lateral flows or groundwater flows), or the water discharged from upstream aquifers into surface water bodies and later contributed to rivers flows through surface pathways, were all accounted as baseflow contribution. The watersheds in Streletskiy et al. (2015) are over a permafrost region and have dual layers of frozen soil (permafrost table and seasonal frozen layer). Obviously, our study region is much less complicated in terms of frozen dynamics and its impact on water flows. However, their findings such as the frozen soil precluding surface water infiltration and building up hydrostatic pressure within the soil column, and the months-long time-lag of late summer precipitation contributing to stream flow in winter, are in support of our model and results. 7. Thanks for this comment. We have enhanced the discussions in Section 5. 8. The description has been revised by including the discussions given in Reply #2 and 3. 9. The main constraint for applying the six USGS methods used in this study is Eq (A1). To the best of our knowledge, improvement of these methods for application over large watersheds is still limited. 10. The modelling hypothesis of this study is not limited by watershed size, but if GRACE data is directly used for estimate TWS, the method could be limited by its large foot print. With the research advance in downscaling GRACE products, it is expected the method could be applied to small watersheds

in the future. We will revise the discussion and provide additional reference.

Reference: Rudra, R., Ahmed, I., Khan, A. A., Singh, K. G., Goel, P. K., Khayer, M., and Dickinson, T.: Use of Baseflow Indices to Delineate Baseflow Dominated and Rapid Response Flow Dominated Watersheds. Canadian Biosystems Engineering 57, 1–11. doi.org/10.7451/CBE.2015.57.1.1, 2015. Wahr, J., Swenson, S., & Velicogna, I.: Accuracy of GRACE mass estimates. Geophysical Research Letters, 33, L06401. https://doi.org/10.1029/2005GL025305, 2006. Wang, S., Huang, J., Li, J., Rivera, A., McKenney, D.W., and Sheffield, J.: Assessment of water budget for sixteen large drainage basins in Canada. Journal of Hydrology, 512, 1-15. doi:10.1016/j.jhydrol.2014.02.058, 2014a Wang, S., McKenney, D.W., Shang, J., and Li, J.: A national scale assessment of long-term water budget closures for Canada's watersheds. Journal of Geophysical Research: Atmospheres, 119, 8712–8725. doi:10.1002/2014JD021951, 2014b. Wang, S., Huang, J., Yang, D., Pavlic, G., and Li, J.: Longterm water budget imbalances and error sources for cold region drainage basins. Hydrological Processes, 29, 2125–2136. doi: 10.1002/hyp.10343, 2015. Wang, S., Zhou, F., and Russell, H.A.J.: Estimating snow mass and peak river flows for the Mackenzie River basin using GRACE satellite observations. Remote Sensing, 9, 256. doi: 10.3390/rs9030256, 2017. Wang, S. and Russell, H.A.J.: Forecasting snowmelt-induced flooding using GRACE satellite data: A case study for the Red River watershed. Canadian Journal of Remote Sensing, 42, 203-213. doi: 10.1080/07038992.2016.1171134, 2016.

[Figure]

Fig. RC2-1: Yearly calibrated baseflow coefficient (k) from our model vs. that inversely
calculated from observations.

**Fig. 1.** Fig. RC2-1: Yearly calibrated baseflow coefficient (k) from our model vs. that inversely
calculated from observations.

**Drainable water variations (excluding SWE) in winter for model construction**

Fig RC2-2: The variation range of drainable water storage used for model calibration

**Fig. 2.** Fig RC2-2: The variation range of drainable water storage used for model calibration

Fig RC2-3: The major watersheds in central-east Canada.

Table RC2-1. The major watersheds in central-east Canada.

| Station ID | Name |
|---|---|
| 04AB001 | HAYES RIVER BELOW GODS RIVER |
| 04CC001 | SEVERN RIVER AT LIMESTONE RAPIDS |
| 04DC001 | WINISK RIVER BELOW ASHEWEIG RIVER TRIBUTARY |
| 04DC002 | SHAMATTAWA RIVER AT OUTLET OF SHAMATTAWA LAKE |
| 04EA001 | EKWAN RIVER BELOW NORTH WASHAGAMI RIVER |
| 04FC001 | ATTAWAPISKAT RIVER BELOW MUKETEI RIVER |
| 04HA001 | ALBANY RIVER NEAR HAT ISLAND |
| 04LG004 | MOOSE RIVER ABOVE MOOSE RIVER |
| 04ME003 | ABITIBI RIVER AT ONAKAWANA |

**Fig. 3.** Fig RC2-3: The major watersheds in central-east Canada.

[Figure]

---

## Author Response (AR2)

Dear Dr. Markus Hrachowitz,

Thank you very much for your comments and providing the references. I fully agree with you and the reviewer regarding the possibility of groundwater inter-watershed flows. Please see my response in more detail below. The manuscript has been revised by addressing all of the three comments, as shown in the track-change version.

We greatly appreciate your time and help on improving our manuscript.

Sincerely,

Shusen Wang and Co-Authors

**Response to reviewer's comments:**

1. I cannot agree with the assertion that Eq (1) does not built on the assumption of watershed water balance closure, although it might not be a big issue for this watershed. I did not find solid evidence for the declaration that the subsurface water change Sg only contributes to the baseflow. How do the authors know there are no groundwater systems across two watersheds?

Authors: We agree with the comment, and yes, watershed water imbalance can affect the model performance. If water balance was an issue, for example, due to the inter-watershed groundwater flows, the model would misrepresent the natural hydrological system, particularly for its baseflow. Our previous studies on water balance (Wang et al., 2014a, b) for Canada's watersheds showed that the Albany watershed had a fairly good water closure, suggesting that the data quality issues and groundwater inter-watershed flows were likely small. Discussions on the potential impact of groundwater inter-watershed flows on our modelling are added in Paragraph 5 of the Discussion Section (Page 18). The studies of Bouaziz et al. (2018) and Hulsman et al. (2021) helped the discussions and were cited therein.

2. As declared as physically-based model, the authors define the meaning of the parameters as shown in Table 1. I understand the authors include the variable Tacc to account for the annual/yearly variations of watershed conditions over the freezing period. Clearly, the parameters are assumed to be constant when using the 15-year winter data for calibration. But I do not understand why the derived parameter K0 derived from the winter period can work for the summer period. It did not consider the variation of K0 for summer periods. Therefore, it should not work well in the summer periods. In fact, we can see this in Fig. 8. In comparison to other methods, it yields much higher baseflow coefficients for the summer periods like in years 2005,2006,2010,2011, and so on. In my understanding, this definitely means a systematic bias over a year. Whereas, Fig 9 easily mislead the readers, since the proposed method originates from the 15-year data with a kind of statistical meaning. Therefore, I can not agree with the authors' explanation. The limitation should be discussed clearly.

Authors: The K0 is defined as the K for summer when frozen soil is absent. The baseflow varies in summer due to the water storage changes, rather than the K changes, in our model. Our model is basically a modified linear reservoir model in summer so K is treated as a constant (K0) (see different model comparisons for the Albany basin in Wang, 2019). In winter, frozen conditions limited liquid water transfers and reduced soil/aquifer hydraulic conductivity, and K becomes dynamic with freezing temperatures. The above-mentioned years had dry summers. Our model estimated little surface runoff for these years in summer. This is not a result of higher baseflow coefficients in these summers. Rather, it is the result that our model estimated the watershed water storage was mainly attributed to subsurface water in these summers. The baseflow coefficients estimated from both our model and inversely calculated from observations (Fig. RC2-1) also did not suggest higher values for these years. Discussions on this was added in Paragraph 7 of the Discussion Section (Page 19) in the revised manuscript.

3. The stream flow data in Fig. 8a is different from that in the other three subplots. There needs some explanation. Please scale it for better comparison with other methods.

Authors: Thanks for the comment! The difference is due to that the "months" in GRACE data are slightly different with the calendar months (See Fig 6 for detail). The "GRACE month" varied from year to year so it is not an ideal option to use it to scale other Q data that were based on calendar month (standard). Luckily, the difference between the "GRACE month" and the calendar month is mostly small and it had minor impact on our analyses. Explanation is added in the Figure caption in the revised manuscript.